# Growth Curves and Body Condition of Young Cats and Their Relation to Maternal Body Condition

**DOI:** 10.3390/ani12111373

**Published:** 2022-05-27

**Authors:** Han Opsomer, Annette Liesegang, Daniel Brugger, Brigitta Wichert

**Affiliations:** Institute of Animal Nutrition and Dietetics, Vetsuisse Faculty, University of Zurich, Winterthurerstrasse 270, 8057 Zurich, Switzerland; hopsomer@nutrivet.uzh.ch (H.O.); dbrugger@nutrivet.uzh.ch (D.B.); bwichert@nutrivet.uzh.ch (B.W.)

**Keywords:** feline obesity, overweight, development, growth patterns, predictive factors

## Abstract

**Simple Summary:**

The aim of this study was to assess the effect of the litter, the individual and the mother on the likelihood that kittens become overweight by 8 months of age. Since efforts to investigate these factors in cats often do not account for variables that increase the likelihood of obesity in adult life (e.g., neutering), data on an intact cat family over 14 years were used. Both males and kittens (regardless of sex) from overweight mothers were more prone to being overweight at 8 months. A greater litter size appeared to protect kittens from becoming overweight, in contrast to a greater birthweight, which seemed to predispose them for it. The growth rate of all kittens differed according to their own (lean vs. overweight and male vs. female) and their mother’s (overweight vs. variable vs. lean) phenotype. In females but not in males, own and maternal phenotype affected the time to reach peak weight as well. Based on the results from this study, birthweight, growth rate and maternal overweight can indicate a predisposition of kittens to become overweight.

**Abstract:**

The aim of the present study was to assess factors like litter, individual and maternal effects on kitten overweight at 8 months of age, defined as body condition score (BCS) ≥ 6, in an intact cat family. To minimize confounding, a homogenized cat population was used. After categorization of the life weight data according to the kittens’ sex, BCS and maternal non-pregnant phenotype (overweight (OM), lean (LM), variable (VM)), analyses including Pearson’s correlation coefficients, two-way ANOVA, linear, linear broken-line regression and repeated measures mixed model analyses were performed. Overweight and OM kittens gained weight most quickly, and females reached their peak weight earlier than males (6.2 ± 0.6 vs. 7.4 ± 0.2 months). In females but not in males the age to reach peak weight differed markedly according to category. Male (5.82 ± 0.09, *p* < 0.01) and OM kittens’ (5.80 ± 0.11, *p* = 0.07) BCS at 8 months was higher and they were heavier than their counterparts, from 3 and 5 months onwards, respectively. Litter size negatively correlated with overweight (r = −0.30, *p* < 0.01) and birthweight showed a positive correlation to live weight (R^2^ = 0.05, *p* = 0.05) and monthly weight gain (R^2^ = 0.18, *p* < 0.01) over time. This study supports routine monitoring of birthweight, growth rate and maternal phenotype prior to pregnancy to identify kittens at risk for becoming overweight.

## 1. Introduction

Due to their high incidence and association with pathologies [1,2,3], being overweight or obese forms a major risk for longevity and life quality in both humans and their pets [2,3,4]. The often unrewarding treatment [5,6] and, at least in cats, the inability or unwillingness of owners to correctly recognize overweight in their pet [7,8,9,10], has led research to be focused on prevention rather than cure.

In cats, earlier studies identified important risk factors like neutering, sex, housing, age, diet, owner interaction, breed and individual behavior [4,6,9,10,11,12,13,14]. Nevertheless, the already-high occurrence of overweight in young cats [9] promotes the need for further research on developmental traits (genetic as well as epigenetic) predisposing cats to obesity, as has been done in humans [3,15,16].

Ideally, such studies would point out easily determinable and useable parameters to identify kittens at risk of becoming overweight. However, with the exception of sex (since males are more at risk [4,7,12,17]), research in growing cats has not yet been able to clearly determine other early life predictive/ influential variables. On the other hand, a relationship between birthweight and a predisposition towards obesity was shown in humans (both offspring with low and high birthweights are predisposed [15,18]), pigs (offspring with low birthweights are predisposed [19]) and dogs (offspring with low birthweights are predisposed [20]). In addition, the influence of litter size on the likelihood to become overweight was pointed out in rodents (small litters are more predisposed [21,22]) and pigs (large litters are more predisposed [23,24]). As the noted associations of birthweight and litter size with overweight differ between species, these findings cannot be extrapolated to cats.

A possible explanation is that most studies on this subject in cats include different factors known to influence obesity like neutering (often mainly the males), diet (no uniform feeding) and restrictive feeding, which can all affect intake and/ or energy expenditure, impairing interpretation of the results [9,17,25,26,27]. As has been advocated in human research [15], studies that exclude possible confounding factors are warranted to distinguish genetic from epigenetic (so-called “thrifty phenotype”) effects on the predisposition towards obesity. Due to the paucity of animal models for human studies on this subject, the use of cats to this end is briefly discussed.

Remarkable in feline studies for maternal factors is that the body condition of the queen prior to pregnancy is often forgotten, insufficiently described, or substituted by body weight, which, although an indicator, cannot be considered the same as body condition [17,28,29]. Using more appropriate parameters could identify effects that may have been overseen before.

It was hypothesized that maternal phenotype (non-pregnant body condition score), litter (size at parturition, size at rearing, mortality within the litter) and individual effects (birthweight and sex) influence the likelihood to become overweight or obese. The present study aimed to assess the factors litter, individual and maternal effects on kitten overweight at 8 months of age, defined as body condition score (BCS) ≥ 6, in an intact cat family. Simultaneously, typically influential factors were reduced or eliminated to achieve a more reliable result.

## 2. Materials and Methods

### 2.1. Animals

Cats belonged to the colony of experimental cats housed according to Swiss animal welfare legislation under permission 144, issued by Canton Zurich, Switzerland.

All cats included in the present investigation belong to a multigenerational, mixed breed cat colony, wherein certain animals showed a genetic predisposition to early-onset obesity [30]. Routinely collected data over a period of 14 years (2007–2021) on 29 litters from a mixture of primi- and multiparous queens, were considered for analysis. Queens were used in several non-invasive experiments like palatability or energy requirement assessment. Breeding never occurred within one year of the queen being included in an experiment. Exclusion of kittens from the data set was based on significant pathology (e.g., orthopedic developmental conditions, feline infectious peritonitis), mortality or departure from the colony prior to the age of 8 months. As adult cats were excluded from the colony when abnormal conditions developed, no additional exclusion criteria were implemented for the queens. In the end, data from 85 kittens from birth until 12 months of age were used.

These were subsequently categorized based on sex (all animals remained intact) in combination with body condition score (BCS) at 8 months of age (<6: lean, ≥6: overweight) or sex in combination with the phenotype of the non-pregnant mother throughout life (BCS always <6: lean mother (LM), BCS always ≥6: overweight mother (OM), BCS differing between < and ≥6: variable mother (VM)). The inclusion of maternal phenotype VM to categorize kittens aimed to widen the phenotypical spectrum, thereby making the study results transferable to field conditions. BCS scoring of kittens (specified below) and adult cats (at least twice yearly) was conducted under the supervision of one of the authors. General body condition in adult cats was assessed weekly by the same veterinarians of the institute. The effect of parity of the mother and type of delivery were not assessed as only a small number of the cats were multiparous and only two parturitions required a caesarean section.

The data set included 48 male and 37 female kittens. The further spread of the population is specified in Appendix A.

The vitality of the kittens was routinely assessed at parturition and thereafter. Kittens were dewormed at 2 and 5 weeks of age with fenbendazole (Panacur^®^, MSD Animal Health GmbH, Luzerne, Switzerland) for three consecutive days (20 mg/kg p.o.). Adult animals only received parasite treatment based on clinical and routine fecal examinations. At 8, 12 and 16 weeks of age, kittens received their immunization against feline panleukopenia virus, calicivirus and herpes virus. At 9 and 13 weeks of age, they were vaccinated against feline leukemia. No remarkable effects on weight gain or general appearance were ever noted as a result of these standard treatments. Vaccinations were repeated yearly.

The cats were housed in groups of two to ten animals depending on behavior and sex. During gestation, the queens stayed in their usual groups until two weeks prior to expected birth. At this time, the queens were isolated from the main colony and housed in the breeding rooms where they remained until weaning when kittens aged 8 weeks. At this point, the queen was reintroduced to the main cat colony. Offspring were grouped according to sex and introduced to the main cat colony at 5 months of age. Artificial light was provided with an automatic timer from 7 a.m. to 7 p.m. during group housing and manually from 8 a.m. to 5 p.m. in the breeding rooms. In addition, ample windows in all rooms allowed exposure to natural sunlight, even when no outdoor access was available (e.g., during cleaning procedures or in the breeding rooms). All enclosures were sufficiently provided with toys, resting places (both ground level and elevated), scratching poles and litter boxes. Enclosures and litter boxes were cleaned daily and washed weekly.

From conception until weaning, queens were fed with commercial dry food for growth and pregnant or lactating queens (ad libitum) and wet food for adult cats (twice daily) which were in accordance with the European Pet Food Industry Federation (FEDIAF) guidelines [31] for each category. Kittens were fed the dry food ad libitum and the canned food twice daily as well. No kittens were ever bottle-fed to support growth.

Adult, non-pregnant cats were fed a commercial dry food for adult maintenance (ad libitum) and wet food for adult cats (once daily) according to FEDIAF guidelines [31].

### 2.2. Data Collection

To ensure healthy development, kittens were weighed at birth, daily for the first two weeks of life, then every second day until weaning at 8 weeks. From weaning onwards, they were weighed on a weekly basis and body condition score (BCS) was determined on a monthly basis using the 9-point scale according to Laflamme [32] that was modified for growing cats by Ghielmetti et al. [33]. For data analysis, however, only birthweight, monthly weight and BCS at 8 months were considered. BCS at 8 months was taken as the reference as all growth curves were expected to have stabilized by this age.

Per group (according to sex and BCS at 8 months or phenotype of the mother), the means and standard deviations were used to visualize growth curves.

Classification of queens was based on continuous monitoring of non-pregnant BCS using the 9-point scale according to Laflamme [32].

### 2.3. Statistical Analysis

The individual animal was the experimental unit for all subsequent procedures, which were conducted with SAS 9.4 (SAS Institute Inc., Cary, NC, USA). The only exceptions were the linear broken-line regression models, where respective arithmetic mean values for sex and phenotype were used to avoid overestimation of the degrees of freedom, due to otherwise imbalance in X and Y coordinates, and the associated loss in the precision of estimation.

Pearson’s correlation coefficients were calculated between litter size at parturition, the number of kittens deceased prior to the age of 12 months, number or reared kittens, birth weight and BCS at 8 months of age with the procedure CORR.

Linear regression models (y = a + bx) were estimated for the effects of birth weight on average monthly body weight and average monthly weight gain, respectively, over the first 12 months of life, applying the procedure REG.

Linear broken line analysis was conducted with the procedure NLMIXED on kittens’ body weight development over the first 12 months of life, with respect to sex and discrimination between lean and overweight kittens. Furthermore, kittens’ body weight was analyzed with respect to the mother’s phenotype and by discriminating between female and male kittens.

General linear models were estimated with the procedure GLM, applying phenotype of the mother, sex, and the interaction as independent variables for the analysis of the number of kittens deceased prior to the age of 12 months, number of reared kittens, birth weight and BCS at 8 months of life. The Tukey–Kramer test was applied to test differences between multiple LSMean values with respect to the Type I error rate.

Repeated measures mixed model analysis was performed with the procedure MIXED to analyze kittens’ monthly body weight and body weight gain, respectively, over the first 8 months of life according to the phenotype of the mother, time, sex of the kitten and interactions (phenotype*time, sex*time), including the individual birth weight as a continuous variable. The individual kitten nested within the mother’s phenotype was set as a random parameter. In each case, a variance component covariance structure was assumed.

For all statistical procedures, *p* ≤ 0.05 was defined as a Type I error threshold.

## 3. Results

The average litter size was 4.5 ± 1.5 kittens, with a loss of 0–2 kittens per litter (median 0) prior to adulthood (data not shown). Means and standard deviations for monthly weight and for calculated daily weight gain per month are shown in Appendix A. Maximal daily weight gain was reached at 3 months for all groups except for overweight males and all OM kittens, which reached their peak in daily weight gain at 4 months of age. The birthweights of the kittens were on average 105 g (range between 69 and 138 g, data not shown).

### 3.1. Relationship of BCS at 8 Months and Maternal Phenotype to Weight Development

#### 3.1.1. General

Kittens which were determined to be overweight at 8 months grew more quickly (685.14 ± 23.80 g/month; *p* < 0.01) and over a longer period of time (statistical breakpoint at 6.88 ± 0.18 months; *p* < 0.01) than kittens lean at 8 months (588.29 ± 16.72 g/month; *p* < 0.01 and statistical breakpoint at 6.55 ± 0.18 g/month; *p* < 0.01). In addition, overweight kittens lost weight (−83.23 ± 30.10 g/month; *p* = 0.02) after reaching their peak life weight, in contrast to lean kittens that plateaued (33.91 ± 21.15 g/month; *p* = 0.13) (data not further shown).

With regard to the phenotype effect of the mother, slopes of weight development over time were lowest for VM kittens (583.68 ± 14.82 g/month; *p* < 0.01) followed by LM (596.76 ± 17.12 g/month; *p* < 0.01) and OM (685.45 ± 24.65 g/month; *p* < 0.01) kittens, with respective peak live weights (weight at statistical breakpoint) being lowest for LM (3873.09 ± 83.60 g; *p* < 0.01) kittens followed by VM (4257.94 ± 73.03 g; *p* < 0.01) and OM (4563.79 ± 90.87 g; *p* < 0.01) kittens. Peak weight (statistical breakpoint) was reached first in LM kittens (6.52 ± 0.19 months; *p* < 0.01), followed by OM (6.80 ± 0.19 months; *p* < 0.01), and VM (7.28 ± 0.15 months; *p* < 0.01) kittens. After reaching their statistical breakpoint, OM and VM kittens lost weight until the end of the observation period (12 months) (−70.64 ± 31.19 g/month; *p* = 0.04 and −176.60 ± 30.38 g/month; *p* < 0.01), whereas LM kittens gained weight (62.93 ± 21.67 g/month; *p* = 0.01) (data not further shown).

#### 3.1.2. According to Sex

The linear broken-line regression of average live weight development over the first 12 months of life per sex (male, female) and BCS at 8 months of the kittens or phenotype of the mother (OM, VM, LM) are shown in Figure 1 (Table 1 shows the statistical measures of the respective models). For each category, males on average reached higher body weights than females. In male kittens, the time point of peak live weight remained constant at 7.4 ± 0.2 months (*p* < 0.1) for all categories, whereas in females it took longer for overweight (6.5 ± 0.5 months; *p* < 0.1) compared to lean (5.7 ± 0.5 months; *p* < 0.1) kittens to reach their maximum, and in increasing order for LM (5.7 ± 0.5 months; *p* < 0.1) to OM (6.22 ± 0.5 months; *p* < 0.1) to VM (7.4 ± 0.4 months; *p* < 0.1) kittens. After the statistical breakpoint, overweight males tended to (−172.67 ± 87.68 g/month; *p* = 0.07) and VM (−215.60 ± 94.50 g/month; *p* = 0.04) and OM males (−96.13 ± 40.34 g/month; *p* = 0.03) significantly lost weight. It should be noted that VM and OM males significantly lost weight after the statistical breakpoint and VM males did so more rapidly than OM males did. In contrast, in the females’ group, only VM females significantly lost weight (−88.10 ± 40.07 g/month; *p* = 0.05) and LM females significantly gained weight (84.67 ± 39.93 g/month; *p* = 0.05) after the statistical breakpoint. All other categories in combination with sex plateaued after peak live weights were reached. The further trend of the slopes in females mimicked those of the general curves. Weight gain in males however increased from LM (662.06 ± 15.55 g/month; *p* < 0.01) to VM (682.91 ± 46.68 g/month; *p* < 0.01) to OM (726.95 ± 19.68 g/month; *p* < 0.01) (Figure 1 and Table 1).

### 3.2. Relationship of Litter Traits and Birthweight with Kitten’s BCS at 8 Months

A positive Pearson correlation was detected for the born and reared litter size (r = 0.89, *p* < 0.01) as well as a negative correlation between reared litter size and mortality within a litter (r = −0.26, *p* = 0.02). Interestingly, both born (r = −0.32, *p* < 0.01) and reared (r = −0.30, *p* < 0.01) litter size showed a weak but significant negative correlation with the kittens’ BCS at 8 months (data not further shown).

Linear regression analysis showed a positive relationship of live weight over time (y = 8.446x + 1990.386, R^2^ = 0.05, *p* = 0.05) and monthly weight gain (y = 4.645x − 49,692, R^2^ = 0.18, *p* < 0.01) with birthweight (data not further shown).

### 3.3. Relationship of Sex and Maternal Phenotype to Litter Characteristics, Birthweight and BCS at 8 Months

The influence of sex and the mother’s phenotype was also evaluated with two-way ANOVA on the number of reared and deceased kittens per litter, kitten birthweight, and kitten BCS at 8 months. These showed no influence of sex or mother’s phenotype on the reared litter size or mortality within the litter (data not shown), respectively. Males were heavier at birth than females (110.09 ± 2.26 g and 103.39 ± 3.06 g, respectively; *p* = 0.08) and generally showed a higher BCS at 8 months of life (5.8 ± 0.1 and 5.4 ± 0.1; *p* < 0.01). With respect to the mother’s phenotype, OM kittens showed a higher BCS at 8 months of life than VM or LM (5.8 ± 0.1 vs. 5.5 ± 0.2 vs. 5.5 ± 0.1 for OM vs. VM vs. LM, respectively; *p* = 0.07) (Figure 2). No effect of maternal phenotype on birthweight was seen (data not shown).

### 3.4. Weight Development according to Repeated Mixed Measures Model in Relation to Sex and Maternal Phenotype

According to repeated measures mixed model analysis, birthweight affected monthly life weight during the first 8 months of life (positive effect; *p* < 0.01) (Figure 3) but not the monthly weight gain (Figure 4). Analysis of monthly life weight showed males to be heavier than females from 3 months (150.88 ± 84.58 g, *p* = 0.07), and this difference increased onwards (Figure 3). OM kittens were heavier than VM and LM kittens (154.40 ± 45.46 g; *p* < 0.01 and 207.70 ± 35.20 g; *p* < 0.01 respectively) (Figure 3), what became apparent at 5 (225.82 ± 124.54 g; *p* = 0.07) and 4 (182.44 ± 97.88 g; *p* = 0.06) months and was first statistically significant at 6 and 5 months (337.63 ± 124.54 g; *p* < 0.01 and 217.54 ± 97.88 g; *p* = 0.02), respectively (data not further shown). The monthly live weight of OM kittens diverged from that of VM and LM kittens until 6 months after which the difference plateaued. Neither in general nor at any given time point was there a significant difference between VM and LM kittens (data not shown).

Analysis of the monthly gain (Figure 4) showed that males gained more than females (193.56 ± 14.85 g/month; *p* < 0.01), but that this first became statistically significant at 3 months of age (113.87 ± 41.28 g/month; *p* < 0.01) (data not further shown). OM kittens gained more than LM kittens did (49.50 ± 17.19 g/month; *p* < 0.01); however, per time point, this was only statistically significant at 4 and 6 months (200.02 ± 47.91 g/month; *p* < 0.01 and 163.86 ± 47.77 g/month; *p* < 0.01) (data not further shown). Compared to VM kittens, OM kittens only gained statistically significantly more in the 4th month (168.28 ± 60.78 g/month; *p* < 0.01) (data not further shown). OM kittens gained increasingly more than LM kittens after weaning until 6 months (with a decrease at 5 months), after which the difference in monthly gain decreased and was inverse in the 8th month (Figure 4). Statistically, VM kittens only gained more than LM kittens in the 7th month (124.91 ± 53.96 g/month; *p* = 0.02 (data not further shown).

## 4. Discussion

In this study, the relationship between maternal phenotype, litter (size at parturition, size at rearing, mortality within the litter) and individual effects (birthweight and sex), and the likelihood of being overweight at 8 months was assessed. The analyzed sample can be considered unique, since the homogenous management of all cats in this population reduced or eliminated possibly confounding effects of multiple factors which have been related to the development of obesity, birthweight or growth rate (e.g., neutering, diet, environment, suckling time, breed, owner characteristics [6,9,10,11,29,34,35]). In addition, the inclusion of the maternal phenotype (lean, variable, or overweight) instead of maternal body weight has, to the authors’ knowledge, never been conducted before.

It is important to consider that even though an effect of neutering was excluded, our data still indicated that in cats, males are generally heavier and more prone to obesity at a young age than females are. Our data, therefore, confirm the results from other studies where this confounding factor was not excluded [4,7,12,17].

The birthweights within the studied population corresponded well to ranges reported elsewhere [17,26,28,29,35]. Males were generally heavier at birth than females, which corresponds to earlier, larger data sets that found higher birthweights for males and kittens from smaller litters [26,29,35]. The sample size is likely even more relevant when breed effects are considered, since some studies reported differences in birthweight according to breed [28,29,35].

Furthermore, our data suggest that kittens with higher birthweights gained weight more quickly. Monthly weight gain in turn was greater in kittens determined to be overweight at 8 months of age, which is in line with earlier findings [25]. In contrast, other authors have indicated that lower kitten birthweight can be associated with more rapid weight gain as well [29]. This discrepancy could, however, be explained by the fact that kittens with a lower birthweight are often bottle-fed [36], which was not the case in the studied population. In humans, the relationship between birthweight and risk to develop obesity shows a J- or U-curve [15], where high birthweight may be a more important risk factor [18]. This may also be the case in felines in contrast to e.g., pigs [19] and dogs [20] where only a low birthweight has been correlated with an increased likelihood to become overweight. This, however, has to be shown by future research. If this analogy between cats and humans could be confirmed, the cat may be an appropriate translational animal model for human obesity research as has been suggested by others [37,38], despite marked physiological differences.

The life weight development of our kittens reached its peak later in males overweight at 8 months and all OM kittens compared to all other respective categories. Since OM kittens could be considered prone to developing obesity (greater weight gain and total weight at the statistical breakpoint compared to VM and LM kittens, respectively, regardless of sex), our data indicate that predisposed cats gain weight more quickly and over a longer period of time. Interestingly, when assessed per sex, the difference in statistical breakpoint according to maternal phenotype and BCS at 8 months was only found in females, whereas the growth curve of all males reached their peak around the same time regardless of their further categorization. Overall, males reached their peak weight later in life compared to female cats. The assessment of the statistical breakpoints therefore not only confirms that males take longer to reach their adult size [27,29,35], but can also contribute to the explanation of the mechanism(s) that predispose(s) cats within this population to obesity, which is further discussed below.

The analysis of growth curves with respect to the mothers’ phenotype suggests a genetic difference as well. This is supported by the fact that within a part of the studied population, a genetic predisposition towards obesity has already been indicated [30]. LM and OM mothers could then be considered two extremes, whereas the VM group would represent a heterogeneous mix. Obesity originates from a continuous positive energy balance (energy intake is greater than energy expenditure). This could be due to reduced satiety mechanisms (increased intake), lowering metabolic rate (reduced expenditure), or both. Although a sex-dependent effect on energy expenditure, as has been found for neutering [12,39], cannot be fully excluded, it seems unlikely to have caused the different statistical breakpoints for females but not males. The predominant role of altered satiety is supported by a previous study with a subset of the population where a difference in food intake but not energy expenditure was shown [33]. It can thus be assumed that a change in satiety mainly causes the noted effects within this cat colony. Since estrogen is known to improve satiety [39,40], a genetically altered effect thereof could explain the shift in slope and, therefore, statistical breakpoint value in females. It would also explain why the time of and weight at the breakpoint for lean and LM females corresponds to the likely age and weight of puberty onset in female kittens [41,42], whereas for the other female categories, the breakpoint occurs later. Since in males, no link between weight and onset of puberty is mentioned in the literature, the similar breakpoints in all the male categories around the time of puberty [42,43] further support the role of sex hormones in the noted effects. If VM kittens would indeed represent a heterogeneous mix of LM and OM, negative feedback (hence weight loss) when a certain satiety signal threshold is reached, would be more pronounced than in OM kittens. This would explain why VM females and VM males lose weight or lose weight more rapidly after the breakpoint, compared to their OM counterparts. It is, however, less clear why the breakpoint of VM females occurs later than that of OM females. Despite this later breakpoint, VM females grew slower than their LM and OM counterparts. Additionally, they did not reach the same value at the breakpoint, compared to the OM females. An effect of other endocrinological influences (e.g., leptin) could therefore play a role as well [39,44].

Since phenotype is determined not only by genetics but epigenetics as well, influential factors on the latter (pre-and postnatal nutrition and environment) need to be considered as well [15,45]. The rehoming at 5 months could for example explain why weight gain differences between phenotypic categories (LM, VM, OM) decreased around this time [46]. Despite efforts to standardize aspects of epigenetics, some differences unavoidably occurred. This can be speculated to be the basis for the weak but significant negative correlation of litter size (both born and reared) with the tendency to become overweight by 8 months, similar to effects in rodents [21,22], but dissimilar to effects in pigs [23,24]. Other influential factors like the effect of year, season, and feed changes could have played a role too. A relatively larger influence of epigenetics on categorization according to maternal phenotype rather than sex, can also argue the occurrence of the significant weight difference for the former at a later age. This is also apparent from the relatively larger variability for monthly weight gain, resulting in fewer significantly different time points. Even studies with markedly larger data sets than ours find first divergences from 5 weeks of age onwards [29,47]. As this is typically the age at which the mother’s milk yield no longer covers the kittens’ requirements in any litter [29], the increasing intake of other food components and the response to this, can drive differences to occur. This is supported by the suggestion that longer suckling time (and therefore less intake of other food) can protect kittens from becoming overweight [34]. This may be analogous to findings in human medicine where the early feeding of formulas predisposes babies for being overweight [48]. Whether the early onset of supportive feeding, the quantity of supportive feed, or both, are relevant requires further research.

In addition, it should be considered that the line between genetic and epigenetic influence is not always clearly separated. Litter size and maternal phenotype can, for example, influence milk yield [49,50] and therefore affect kitten development. It would thus be interesting to see if, for example, prospective studies with restricted feeding of spontaneously obese queens or inducing overweight in lean queens (altering phenotype but not genotype) would still render the same effect on the kittens’ growth curves. Such studies can also pinpoint the effect of pre-and postnatal programming [15].

Although studies have found an effect of birth season on development in cats [17], cattle [51], sheep [52], humans [53] and a wide range of other species [45], the present data set was not deemed appropriate to assess this within our population. It should, however, be considered that season and year of birth were indirectly accounted for in our analysis by the inclusion of the individual animal as a random factor for the repeated measures mixed model analysis.

A disadvantage of this study is that only BCS at 8 months was considered. The rationale was that the growth curves of all kittens would have stabilized by this time point, and BCS at 8 months was available for more kittens, resulting in a much larger data set. Although this time point is still a valuable reference (most cats are considered fully grown), it is clear from the linear broken-line regressions (Figure 1) that further stabilization of weight and potentially BCS could still occur. As the risk of obesity is higher during adult life, the use of older reference ages and/or correlation between measured body condition scores at different life stages as done in other studies [17,25] could therefore be beneficial.

In addition, caution is warranted when interpreting data on highly heterogeneous groups like the VM groups in the present study. Separate evaluation of such groups nevertheless holds merit to widen the phenotypical spectrum, thereby transferring study results to field conditions.

The present study advocates the monitoring of maternal phenotype prior to pregnancy, litter size, and birthweight in future research, to render threshold criteria and motivate intervention in practice (e.g., restricted feeding to avoid being overweight from a young age). In addition, the different growth patterns according to sex advocate the inclusion of hormonal balance in future studies to elucidate the (patho-)physiological mechanisms. As the individual relevance of genetic and epigenetic effects remains unclear, prospective studies to distinguish them from one another are warranted as well, in order to promote progress in direction of animal-individual nutrition. As the above-mentioned factors have not yet been adequately researched, Figure 5 depicts the growth curves of lean males and females within this studied population as a practical reference for ideal growth in male and female cats with an expected ideal adult weight at twelve months of 4486 ± 595 g and 3455 ± 614 g respectively. Finally, future research should further address the potential analogy between cats and humans regarding early life effects and the predisposition to obesity, in order to clarify whether felines could be appropriate translational models in this regard. Similarities noted in human studies [15,18,48,54] are, however, promising.

## 5. Conclusions

The present study showed female and male cats determined to be overweight at 8 months to respectively gain weight for longer and more quickly than their lean counterparts. Compared to females, males were more predisposed to becoming overweight (based on BCS at 8 months), even when intact. An investigation of whether BCS at 8 months forms the best reference to predict overweight in adult life is, however, warranted. Based on the results of the present study, it is recommended to monitor and evaluate maternal phenotype prior to pregnancy, litter size and birthweight in future research projects and breeding programs to determine kitten propensity to be overweight. Although further research is warranted, the growth curves illustrated in Figure 5 can cautiously be used as a reference for ideal growth.

## Figures and Tables

**Figure 1 animals-12-01373-f001:**
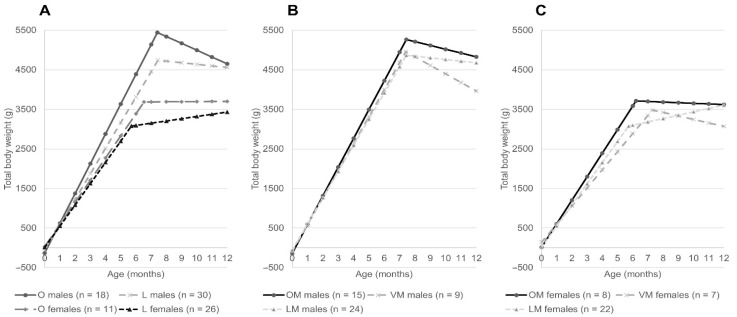
Linear broken-line regressions based on monthly averaged body weight (see Appendix A), according to (**A**) sex and body condition score (BCS) at 8 months, (**B**) for males according to maternal phenotype, (**C**) for females according to maternal phenotype (see Table 1 for detailed information on the statistical measures of the respective regression models). Per category, *n* is given between brackets in the legend. O = kitten BCS ≥ 6 at 8 months, L = kitten BCS < 6 at 8 months, OM = maternal BCS always ≥6, VM = maternal BCS differing between < and ≥6, LM = maternal BCS always <6.

**Figure 2 animals-12-01373-f002:**
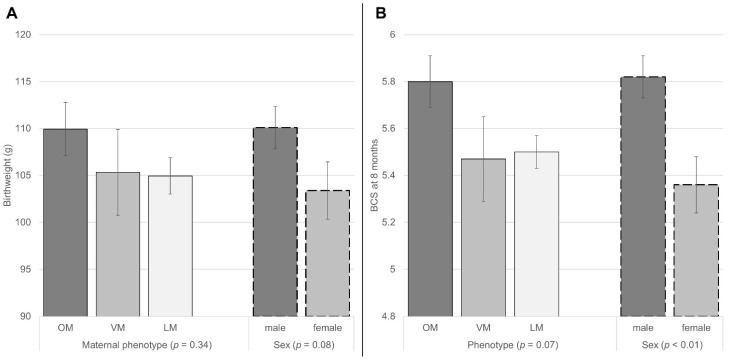
Two-way ANOVA analysis results with independent variables: sex and phenotype, dependent variables: (**A**) birthweight and (**B**) body condition score (BCS) at 8 months (used observations: 85 for both). Columns and brackets show the least square means and their standard errors respectively. Respective *p*-values are depicted per categorization group. *p* < 0.05 was considered significant. ANOVA results for dependent variables: mortality within the litter, and the number of reared kittens are not shown due to a lack of trends or significance. The interaction of phenotype and sex was always at *p* ≥ 0.41.

**Figure 3 animals-12-01373-f003:**
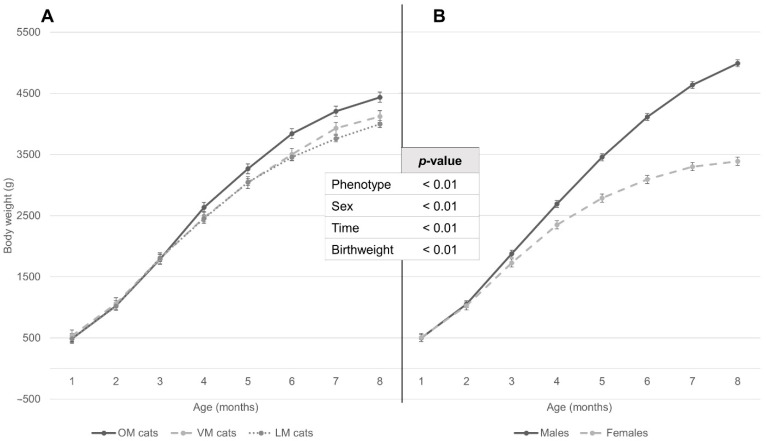
Least square means of total weight development from first to the eighth month of age, according to (**A**) maternal phenotype and (**B**) sex (data from repeated measures mixed model). Standard errors are depicted with brackets per month, for each group. *p*-values for each independent variable used in the repeated measures mixed model are specified in the central table. Both (**A**) and (**B**) are depicted according to the same scale. OM = maternal BCS always ≥6, VM = maternal BCS differing between < and ≥6, LM = maternal BCS always <6. Time significantly interacted with phenotype and sex (both *p* < 0.01).

**Figure 4 animals-12-01373-f004:**
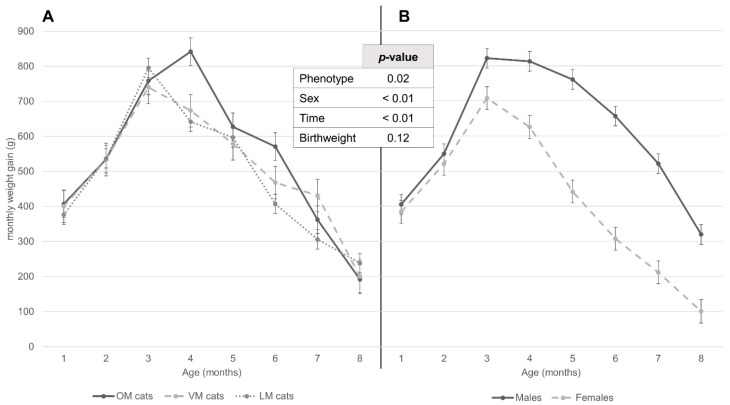
Least square means of monthly weight gain from first to eight months of age, according to (**A**) maternal phenotype and (**B**) sex (data from repeated measures mixed model). Standard errors are depicted with brackets per month, for each group. *p*-values for each independent variable used in the repeated measures mixed model are specified in the central table. Both (**A**,**B**) are depicted according to the same scale. OM = maternal BCS always ≥6, VM = maternal BCS differing between < and ≥6, LM = maternal BCS always <6. Time significantly interacted with phenotype and sex (both *p* < 0.01).

**Figure 5 animals-12-01373-f005:**
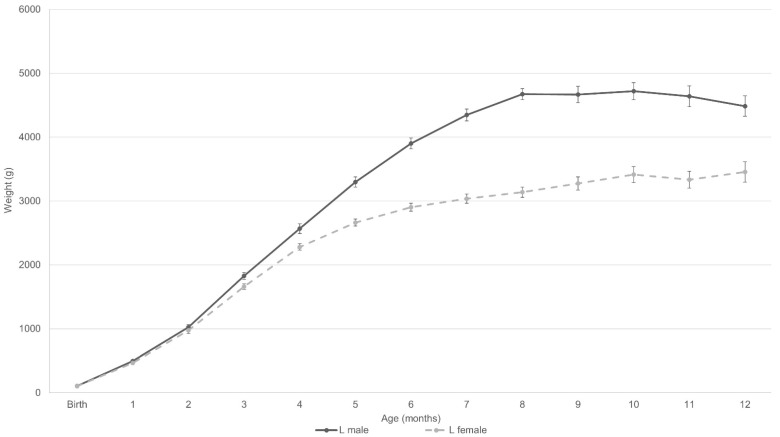
Mean live weight of lean males (L male, n = 30) and lean females (L female, n = 26) from birth until the twelfth month of age. Standard errors are depicted with brackets per month, for both sexes. The illustrated growth curves can cautiously be used as a reference for male and female cats with an expected ideal adult weight at twelve months of 4486 ± 595 g and 3455 ± 614 g respectively.

**Table 1 animals-12-01373-t001:** Linear broken-line regression equations based on monthly averaged body weight (see Appendix A), (**A**) according to sex and body condition score (BCS) at 8 months, (**B**) for males according to maternal phenotype, (**C**) for females according to maternal phenotype (see Figure 1 for graphical depiction).

	Regression Model	Parameter Estimates	*p* Values	R^2^
**A**				
**O males**	Y = −134.07 + b_1_x for x ≤ X_B_	X_B_, 7.40 ± 0.35	<0.01	0.99
	* Y = 6722.70 + b_2_x for x > X_B_	Y_B_, 5444.32 ± 214.44	<0.01	
		b_1_, 753.47 ± 43.30	<0.01	
		* b_2_, −172.67 ± 87.68	0.07	

**L males**	Y = −64.66 + b_1_x for x ≤ X_B_	X_B_, 7.44 ± 0.17	<0.01	1.00
	Y = Y_B_ for x > X_B_	Y_B_, 4740.02 ± 88.98	<0.01	
		b_1_, 645.86 ± 15.77	<0.01	

**O females**	Y = −64.66 + b_1_x for x ≤ X_B_	X_B_, 6.53 ± 0.49	<0.01	1.00
	Y = Y_B_ for x > X_B_	Y_B_, 3685.44 ± 199.75	<0.01	
		b_1_, 560.00 ± 46.43	<0.01	

**L females**	Y = 29.76 + b_1_x for x ≤ X_B_	X_B_, 5.71 ± 0.51	<0.01	0.99
	Y = Y_B_ for x > X_B_	Y_B_, 3077.78 ± 173.91	<0.01	
		b_1_, 537.60 ± 55.17	<0.01	
**B**				
**OM males**	Y = −140.98 + b_1_x for x ≤ X_B_	X_B_, 7.44 ± 0.18	<0.01	1.00
	Y = 5979.85 + b_2_x for x > X_B_	Y_B_, 5264.98 ± 104.83	<0.01	
		b_1_, 726.95 ± 19.68	<0.01	
		b_2_, −96.13 ± 40.34	0.03	

**VM males**	Y = −99.35 + b_1_x for x ≤ X_B_	X_B_, 7.40 ± 0.39	<0.01	0.99
	Y = 6551.24 + b_2_x for x > X_B_	Y_B_, 4955.41 ± 219.05	<0.01	
		b_1_, 682.91 ± 46.68	<0.01	
		b_2_, −215.60 ± 94.50	0.04	

**LM males**	Y = −56.36 + b_1_x for x ≤ X_B_	X_B_, 7.44 ± 0.17	<0.01	1.00
	Y = Y_B_ for x > X_B_	Y_B_, 4866.72 ± 87.88	<0.01	
		b_1_, 662.06 ± 15.55	<0.01	
**C**				
**OM females**	Y = 11.71 + b_1_x for x ≤ X_B_	X_B_, 6.22 ± 0.49	<0.01	1.00
	Y = Y_B_ for x > X_B_	Y_B_, 3713.65 ± 227.96	<0.01	
		b_1_, 595.10 ± 50.65	<0.01	

**VM females**	Y = 138.10 + b_1_x for x ≤ X_B_	X_B_, 7.30 ± 0.27	<0.01	0.99
	Y = 4126.60 + b_2_x for x > X_B_	Y_B_, 3483.33 ± 103.35	<0.01	
		b_1_, 458.15 ± 19.55	<0.01	
		b_2_, −88.10 ± 40.07	0.05	

**LM females**	Y = 9.52 + b_1_x for x ≤ X_B_	X_B_, 5.72 ± 0.54	<0.01	1.00
	Y = 2590.93 + b_2_x for x > X_B_	Y_B_, 3075.42 ± 182.50	<0.01	
		b_1_, 535.81 ± 56.21	<0.01	
		b_2_, 84.67 ± 39.93	0.05	

* Equation based on trend, *p* value not significant. O = kitten BCS ≥ 6 at 8 months, L = kitten BCS < 6 at 8 months, OM = maternal BCS always ≥ 6, VM = maternal BCS differing between < and ≥ 6, LM = maternal BCS always < 6. X_B_ = age (months) at the breakpoint (BP), Y_B_ = weight (g) at the breakpoint (BP), b_1_ = slope prior to BP, b_2_ = slope after BP. *p* < 0.05 was considered significant.

## Data Availability

The analyzed data are available at request from the corresponding author.

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
