# Peer review of "Growth Curves and Body Condition of Young Cats and Their Relation to Maternal Body Condition"

_animals, 2022, doi:10.3390/ani12111373_

Round 1

Reviewer 1 Report

Dear authors, thank you for sharing your work on early onset obesity in felines. Herewith my comments/suggestions for improvement

R11: maybe replace "therefore" by "For that reason" and move this to the beginning of the sentence

R13: "their own" instead of "their"

R14-15: I do not see the relevance of that sentence. The next sentence is the main answer to the research question, so that one should not start with nevertheless and should not be preceeded by a not so important finding

Please formulate a clear aim that should be stated somewhere in the the simple summary, at the beginning of the abstract and at the end of the introduction.

R31: please give the r or r2 that corresponds with the p-values 

R48-49: please re-word, as this sentence is unclear to me

R49-52: Please elaborate more on the findings in these studies and also compare your findings with those studies in the discussion section

Material and methods: I am having difficulties with the variable mother group. I would suggest to leave them out, or to group the cats/kittens from this group to either LM or OM depending on the mother's BCS at the time of pregnancy. This would make the whole manuscript clearer and easier to read. You might even consider only to have a lean (BCS lower or equal to 5) and overweight (BCS of 7 or higher) mother group, as these are generally accepted definitions for lean and obese, and leave the overweight (BCS = 6) out.

R151: either use mean+-sd or use median and range. Please give the median for loss of kittens as well.

R306-307: please mention here again that your kittens were not bottle fed.

R313-325: please explain the weight loss in some cats after reaching peak body weight

R335: please reword "This even more so" as it is unclear

R388: this is indeed a major limitation and it would be nice to look at BCS later in adult life, as it is know that obesity risk is highest in adulthood. Maybe you can elaborate a bit more on this and give more justification why it is still important to look at BCS at 8 months of age (i.e. the age at which domestic shorthairs reach the end of the growth period) (e.g. overweight/obesity at the end of the growth period increases the risk of overweight/obesity later in life).

I do not agree with discussion and conclusion on the VM group, so I strongly suggest to take that group out

Author Response

Reviewer 1:

Dear authors, thank you for sharing your work on early onset obesity in felines. Herewith my comments/suggestions for improvement

Answer: Thank you for your evaluation and your comments which we will answer in the following paragraphs.

  1. Remark: R11: maybe replace "therefore" by "For that reason" and move this to the beginning of the sentence
    Answer: We agree with the reviewer, this makes the sentence more readable. In accordance with remark 4, the simple summary was adapted in such a way that this no longer poses a problem (line 9).

  2. Remark: R13: "their own" instead of "their"
    Answer: We agree with the reviewer, this makes the sentence more readable. We therefore adapted the manuscript accordingly (line 16).

  3. Remark: R14-15: I do not see the relevance of that sentence. The next sentence is the main answer to the research question, so that one should not start with nevertheless and should not be preceeded by a not so important finding.
    Answer: We thank you for the input. We find the sentence regarding the growth patterns relevant as we consider it a major finding that the differences between the growth curves according to predilection for overweight (different categories) are dissimilar when assessed per sex. We do however understand that our other findings should be mentioned first and have adapted the manuscript accordingly (line 13-18).

  4. Remark: Please formulate a clear aim that should be stated somewhere in the simple summary, at the beginning of the abstract and at the end of the introduction.
    Answer: We adjusted the text so that the aim is more clearly stated in the simple summary, the abstract and the introduction, in accordance with the remark (line 9-10, line 20-22, line 74-78).

  5. Remark: R31: please give the r or r2 that corresponds with the p-values 
    Answer: Thank you for the remark. The corresponding R2- values have now been added to the abstract (line 31-32).

  6. Remark: R48-49: please re-word, as this sentence is unclear to me
    Answer: The purpose was to bring across that sex is currently the only early-life risk factor agreed upon to have an influence. Other studies hint towards other variables (e.g. season) but these can be debated. We do agree that this could be described more clearly. We adapted the sentence accordingly (line 49-51).

  7. Remark: R49-52: Please elaborate more on the findings in these studies and also compare your findings with those studies in the discussion section.
    Answer: As noted relations differ between species (for birthweight: in humans both high and low birthweight have been linked to the risk to develop obesity, whereas only the latter has been in dogs and pigs/ for litter size: contrasting findings between rodents and pigs), to elaborate further would cause too much speculation. We however agree that it would be useful to readers to be aware of this. We therefore made a small addition to the text (line 51-58).
    Small additions to the discussion have been made to point this out further (line 337-339 & line 390-392)

  8. Remark: Material and methods: I am having difficulties with the variable mother group. I would suggest to leave them out, or to group the cats/kittens from this group to either LM or OM depending on the mother's BCS at the time of pregnancy. This would make the whole manuscript clearer and easier to read. You might even consider only to have a lean (BCS lower or equal to 5) and overweight (BCS of 7 or higher) mother group, as these are generally accepted definitions for lean and obese, and leave the overweight (BCS = 6) out.
    Answer: We agree that the manuscript would be more comprehensible when only two maternal phenotypes (OM and LM) would be included. On the other hand, categorisation of the queens was based on whether or not they had and maintained a BCS 6, rather than how much they exceeded this often-used threshold for overweight (line 97-98). In addition, as mentioned in the discussion (line 357-358), we suspect (a) genetic factor(s) to play a part in the phenotypical differences of the queens. It would therefore be incorrect to subdivide this variable group among the two other phenotypes.
    We also feel that including this variable group and widening the phenotypical spectrum of the queens, transfers the results of this study to field conditions.
    Furthermore, as the current categorization based on maternal phenotype allows clear comparison of the always overweight and the always lean group, the inclusion of the variable group does not impair interpretation of the results. For this reason, we kept the VM group in the manuscript.
    To make the points clearer to the reader, we included more information (line 428-431) and considered also your comment regarding the discussion. We, by no means, proclaim to have all the answers as much still needs further investigation. We tried to adequately indicate this in the manuscript.

  9. Remark: R151: either use mean+-sd or use median and range. Please give the median for loss of kittens as well.
    Answer: We adapted the manuscript as requested (line 177)

  10. Remark: R306-307: please mention here again that your kittens were not bottle fed.
    Answer: We adapted the manuscript as requested (line 334)

  11. Remark: R313-325: please explain the weight loss in some cats after reaching peak body weight
    Answer: Thank you for that comment. The weight loss after the breakpoint was unexpected. It appears that some later stabilization of body weight occurs. Why this could be different is addressed later (line 376-380). This also caused us to point out that the sole analysis with the BCS at 8 months might be suboptimal (line 421-428). We feel as our theory is already rather speculative, further elaboration in the manuscript would not be beneficial. Clearly, more work is needed to understand the unexpected and yet not understood observations, although the effect appeared in the majority of animals in the different groups mentioned.

  12. Remark: R335: please reword "This even more so" as it is unclear
    Answer: We agree the current phrasing could cause some misinterpretation. We have adapted the manuscript accordingly (line 366-368).

  13. Remark: R388: this is indeed a major limitation and it would be nice to look at BCS later in adult life, as it is know that obesity risk is highest in adulthood. Maybe you can elaborate a bit more on this and give more justification why it is still important to look at BCS at 8 months of age (i.e. the age at which domestic shorthairs reach the end of the growth period) (e.g. overweight/obesity at the end of the growth period increases the risk of overweight/obesity later in life).
    Answer: Thank you for your valid remark. We adapted the manuscript slightly to include this in our discussion (line 420-427).

  14. Remark: I do not agree with discussion and conclusion on the VM group, so I strongly suggest to take that group out.
    Answer: We would like to refer to our answer to remark 8. Please consider line 428-431 where we briefly warn the reader and argument why we continue to include the VM-group.

Reviewer 2 Report

The authors here investigated the growth curves and body condition of kittens and their relation to maternal body condition in a homogenized, non-neutered cat population using statistical methods. The study and the results exposed are interesting and they emphasize the importance of the maternal phenotype, and the birthweight monitoring. The manuscript needs to be revised a little bit by adding more details about the study population. You will find my comments below.

Comments:

Line 73. Were the queens previously used for experiments? And if yes, were those invasive experiments? And how much did you wait after the end of the experiments before breeding the cats? These details should be included if available.

Line 75. Please include details about vaccination status of the animals, de-worming and ecto-parasites treatments, and whether these treatments had an effect on the animals’ weights (through stress for example).

Line 78. Did the inclusion-exclusion criteria concern the kittens only or both the queens and kittens?

Line 78. Was there a difference between the primi- and multiparous queens litter sizes?

Line 79. Besides pathologies, did the authors take into account the parturition method and possible complications? Were the neonatal reflexes evaluated and taken into consideration in the inclusion-exclusion criteria?

Line 81. Once the cats reached 12 months old, did you perform a check up to assess any disorder that could have an interference with their growth and that couldn’t be assessed before puberty or at a younger age?

Line 85. “Throughout life”, what was the frequency of weight and BCS assessment of the mothers?

Line 90. Same as line 75 comment.

Line 113. BCS scoring can be subjective and more information on who did the scoring and whether or not it was the same veterinarian/technician who did it should be added in the materials and methods section

Line 115. Why were the birthweight and BCS at 8 months only considered? What about the other months?

Lines 151-152. Was there a relationship between maternal phenotype and the number of kittens lost?

Line 189. Please correct to “significantly lose weight”

Author Response

Reviewer 2:

The authors here investigated the growth curves and body condition of kittens and their relation to maternal body condition in a homogenized, non-neutered cat population using statistical methods. The study and the results exposed are interesting and they emphasize the importance of the maternal phenotype, and the birthweight monitoring. The manuscript needs to be revised a little bit by adding more details about the study population. You will find my comments below.

Answer: Thank you for your evaluation and your comments which we will answer in the following paragraphs.

  1. Remark: Line 73. Were the queens previously used for experiments? And if yes, were those invasive experiments? And how much did you wait after the end of the experiments before breeding the cats? These details should be included if available.
    Answer: Thank you for this remark. The details are indeed available and are now incorporated in the manuscript (line 86-89).

  2. Remark: Line 75. Please include details about vaccination status of the animals, de-worming and ecto-parasites treatments, and whether these treatments had an effect on the animals’ weights (through stress for example).
    Answer: As requested, we have now included more detail in this regard (line 106-113).

  3. Remark: Line 78. Did the inclusion-exclusion criteria concern the kittens only or both the queens and kittens?
    Answer: As the studied population comprised cats solely kept for research purposes, the exclusion criteria only concerned kittens (line 89-91). As soon as cats developed conditions impairing their use in further experiments they are excluded from the colony (when possible by rehoming). We do agree that should be mentioned and have therefore added it to the manuscript (line 91-93).

  4. Remark: Line 78. Was there a difference between the primi- and multiparous queens litter sizes?
    Answer: We were not able to assess this, as there were only a few multiparous queens. We have now pointed this out in the manuscript (line 101-103).

  5. Remark: Line 79. Besides pathologies, did the authors take into account the parturition method and possible complications? Were the neonatal reflexes evaluated and taken into consideration in the inclusion-exclusion criteria?
    Answer: As there were only two recorded parturitions that required a caesarean section, we considered our data set unsuitable to assess the effect of delivery type. As this is however a valid question, we have now clarified this in the manuscript (line 101-103). Vitality of the kittens was indeed assessed routinely (newly added on line 106) but not directly used as exclusion criteria. Nevertheless, since our data only comprised normal developing kittens (i.e. not requiring additional care; line 89-93), kittens with a reduced vitality that could have impaired development have been excluded.

  6. Remark: Line 81. Once the cats reached 12 months old, did you perform a check up to assess any disorder that could have an interference with their growth and that couldn’t be assessed before puberty or at a younger age?
    Answer: Cats were routinely checked during their lifetime. As mentioned above: as soon as cats developed conditions impairing their use in further experiments they are excluded from the colony (when possible by rehoming). This also concerned younger cats (e.g. two kittens with orthopedic issues that were noticed in adulthood). We do agree some clarification in this regard would be useful and feel this is sufficiently covered by our previously mentioned addition to the manuscript (line 89-93).

  7. Remark: Line 85. “Throughout life”, what was the frequency of weight and BCS assessment of the mothers?
    Answer: The adult cats in our colony are weighted weekly. BCS is assessed at the same time by doctoral students and at least twice a year under the supervision of one of the authors. It should be mentioned that this supervisor has remained the same person over the past 14 years, ensuring a certain continuity in the assessment. As we agree that some specification is warranted, we added a small remark to the text (line 98-101).

  8. Remark: Line 90. Same as line 75 comment.
    Answer: we would like to refer to our answer to remark 2.

  9. Remark: Line 113. BCS scoring can be subjective and more information on who did the scoring and whether or not it was the same veterinarian/technician who did it should be added in the materials and methods section
    Answer: We agree that this is important information to convey to the reader and have added this to our manuscript (line 98-101).

  10. Remark: Line 115. Why were the birthweight and BCS at 8 months only considered? What about the other months?
    Answer: With regard to the weight, we would like to point out that data for every month was indeed used (please see under Data collection, line 139-140). With regard to BCS, only the score at 8 months was used as by this point, we expected all growth curves to have stabilized. Analyses of the scores prior to completion of growth seemed of little value, as growing animals in general only show an increased BCS in extreme cases. We have clarified this in the data collection section as well (line 140-141).
    In our discussion (line 420-427), we do indicate that a supervised assessment at a later age would be valuable as well.

  11. Remark: Lines 151-152. Was there a relationship between maternal phenotype and the number of kittens lost?
    Answer: Even though we agree this could be interesting information, this relation was not assessed.

  12. Remark: Line 189. Please correct to “significantly lose weight”
    Answer: We have adapted the text accordingly (line 215-216).

Round 2

Reviewer 1 Report

Thank your for addressing most of my concerns. I still find the inclusion of the VM group tricky and confusing. If the editor is OK with leaving the VM group in, I would like to see more explanation about why this group was included in the materials and methods

Seems like references 34 and 35 are similar, as 34 is a conference abstract of the publication of 35, I would suggest lo leave reference 34 out.

Author Response

Dear Reviewers,

We thank you for your valued input towards improving this manuscript. We implemented the last latest corrections as requested (please see the respective specifications below).
All revisions were made in the downloadable file from the "Manuscript Information Overview" tab, and were marked as requested.

We look forward to your reply and thank you for all considerations.

Remark: Thank you for addressing most of my concerns. I still find the inclusion of the VM group tricky and confusing. If the editor is OK with leaving the VM group in, I would like to see more explanation about why this group was included in the materials and methods
Answer: As was requested, we included a brief notion in the materials and methods section of the manuscript (line 99-101).

Remark: Seems like references 34 and 35 are similar, as 34 is a conference abstract of the publication of 35, I would suggest lo leave reference 34 out.
Answer: Thank you for the valid remark. Both references were initially included due to their slight differences in the conclusions based on dissimilar classification. We however agree, that this is not essential to this manuscript and have therefore followed the advice to take reference to the conference abstract out.